

# Modeling the short-term fire effects on vegetation dynamics and surface energy in Southern Africa using the improved SSiB4/TRIFFID-Fire model

**Huilin Huang[1*], Yongkang Xue[1,2], Ye Liu[1], Fang Li[3], and Gregory Okin[1]**

5      1 Department of Geography, University of California, Los Angeles, CA 90095, USA

2 Department of Atmospheric & Oceanic Sciences, University of California, Los Angeles, CA 90095, USA

3 International Center for Climate and Environmental Sciences, Institute of Atmospheric Physics, Chinese Academy of Sciences, Beijing, China

10     *Corresponding to: Huilin Huang (hhllbao.work@gmail.com)





**Abstract**

Fire causes abrupt changes in vegetation properties and modifies flux exchanges between land and atmosphere at subseasonal to seasonal scales. Yet these short-term fire effects on vegetation dynamics and surface energy balance have not been comprehensively investigated in the vegetation model coupled with the fire module. This study applies the SSiB4/TRIFFID-Fire model to study the short-term fire impact in Southern Africa with comprehensive evaluations of simulated fire regimes, vegetation productivity, and surface fluxes. We find an annual average reduction in grass cover by 4-8 % for widespread areas between 5-20 °S and a tree cover reduction by 1 % at the southern periphery of tropical rainforests. The fire effects on regional scales accumulate during June-October and peak in November, the beginning of the rainy season. After the fire season ends, the grass cover quickly returns to unburned conditions before the next fire season, while the tree fraction hardly recovers in one rainy season. The vegetation clearance by fire has reduced the leaf area index (LAI) and gross primary productivity (GPP) by 3-5 % and 5-7 % annually, respectively. The exposure of bare soil has enhanced surface albedo and therefore decreased the absorption of shortwave radiation. Annual mean sensible heat has dropped by 1.4 W m$^{-2}$ while the latent heat reduction is small (0.1 W m$^{-2}$) due to the compensating effects between canopy transpiration and soil evaporation. A slight warming effect is simulated after fire, which could be enhanced when the surface darkening effect is incorporated.



## 1. Introduction

Fire is an integral component of the Earth's ecosystem (Bond et al., 2005; Bowman et al., 2009). Through prevalent disturbance on surface biophysical properties (i.e., albedo and vegetation characteristics), fire alters radiative forcing on Earth's surface and modifies the energy flux exchanges between land and atmosphere (Chambers and Chapin, 2002; Bond-Lamberty et

al., 2009). The change in boundary layer may interact with monsoon cycles and negatively influence the monsoonal precipitation (Wendt et al., 2007; De Sales et al., 2016; Saha et al., 2016). Meanwhile, fires also alter atmospheric biogeochemical processes through the release of greenhouse gases (GHGs), aerosols, and volatile organic compounds (VOCs) (Scholes et al., 1996; Li et al., 2019a), exerting radiative forcing in the climate system through GHGs effects,

aerosol-radiation interactions, and aerosol-cloud interactions (Ward et al., 2012; Jiang et al., 2016; Hamilton et al., 2018; Zou et al., 2020).

Fire models have been developed within Dynamic Global Vegetation Models (DGVMs) to explicitly describe the burned area, carbon emission, and fire disturbance on vegetation (Thonicke et al., 2001; Venevsky et al., 2002; Arora and Boer, 2005; Thonicke et al., 2010; Li et

al., 2012; Pfeiffer et al., 2013; Lasslop et al., 2014; Rabin et al., 2018; Burton et al., 2019; Huang et al., 2020b). The fire-coupled DGVMs have been widely used to study the role of fire on vegetation distribution (Bond and Midgley, 2012; Seo and Kim, 2019), terrestrial carbon budget (Li et al., 2014; Yue et al., 2015; Lasslop et al., 2020), surface energy balance (Li et al., 2017; Huang et al., 2020b), and water cycle (Li and Lawrence, 2017). These studies are referred to as

the "long-term fire effects" where the simulations with fire are compared with reference simulations representing "a world without fire." While the long-term fire effects are comprehensively assessed in multi-model studies (Lasslop et al., 2020), the short-term fire



effects at monthly to annual scales have not been quantified in current fire-vegetation models. Every year in the dry season, fire consumes surface vegetation and brings abrupt changes in local

surface properties (albedo and roughness) and energy balance, therefore modifying the land-atmosphere interactions and even interacting with the monsoon in the following wet seasons (De Sales et al., 2016; Saha et al., 2016). These short-term effects are significant at local and regional scales, yet by our knowledge, they have not been assessed in any fire-vegetation model studies. The simulated fire impacts can be compared with those from satellite observations to evaluate

the description of fire-vegetation interactions in DGVMs, bridging the gap between observational and modeling studies.

The short-term fire effects are highly variable among different ecosystems. In this study, we start with fire effects in tropical savannas which have the largest stretch of burned area and the most representative fire-vegetation-climate feedbacks among all the ecosystems (Staver et al.,

2011). Satellite observational studies have quantified the short-term fire effects on albedo change and surface radiation in tropical savanna (Beringer et al., 2003; Veraverbeke et al., 2012; Gatebe et al., 2014; Lopez-Saldana et al., 2015; Saha et al., 2016; Dintwe et al., 2017; Saha et al., 2017; Liu et al., 2019b; Saha et al., 2019). An immediate reduction in surface albedo is reported after fire, associated with ash and charcoal deposition (Govaerts et al., 2002; Myhre et al., 2005).

Some observations found the surface darkening lasted for 10-60 days, followed by a gradual brightening when charcoals were removed by wind or runoff and bare soil was exposed (Lyons et al., 2008; Samain et al., 2008; Gatebe et al., 2014; Saha et al., 2017; Saha et al., 2019). According to recent estimates in Saha et al. (2019), the average albedo anomaly in the year following fire is $+6.51\times10^{-4}$ for all of sub-Saharan Africa (SSA), representing a negative

radiative forcing dominated by surface brightening effects. On the other hand, some studies





found the darkening maintained for more than 4 months, with the brightening effects only in limited areas (Jin and Roy, 2005; Dintwe et al., 2017). Dintwe et al. (2017) reported a positive radiative forcing by 0.18 W m$^{-2}$ averaged over SSA. The variations come from various aspects, including the background climate, soil properties of study regions, burning seasons, and criteria

used to define the "control" pixel (Dintwe et al., 2017; Saha et al., 2019).

Fire effects on savanna vegetation have been widely investigated on site-level studies, which show that vegetation can recover to unburned conditions within 8 days to 12 months after fire (Araújo et al., 2017; das Chagas and Pelicice, 2018). However, fire-vegetation interactions on the regional scale remain unclear, both in the observational and fire modeling studies.

Observational data analysis proposed that fire-caused vegetation loss was an important component in the negative feedback loop between fire and precipitation in Africa, in which fire suppressed precipitation, thereby reducing fuel load and fire in the subsequent season (Saha et al., 2016). The proposed feedbacks are also tested in regional modeling studies which showed that post-fire land condition deterioration resulted in a decrease in wet season rainfall, associated with

atmospheric cooling and subsidence (De Sales et al. 2018), as well as a weakening of West African monsoon progression (De Sales et al., 2016).

This study makes the first attempt to simulate the short-term fire effects on vegetation and surface energy using a fire-coupled dynamic vegetation model. The process-based fire-vegetation model, SSiB4/TRIFFID-Fire, explicitly simulates the burned area, fire disturbance on

vegetation properties, and the subsequent impact on surface energy. The model has been comprehensively evaluated with observed burned area and fire emissions on the global scale and is shown to capture the fire-vegetation interactions under the current climate (Huang et al., 2020b). This study further improves the SSiB4/TRIFFID-Fire to better describe the temporal





variations of fire regimes and vegetation productivity on monthly scales. The model

improvement and experimental design are given in Sect. 2. After comprehensive validation of

the fire-vegetation model performance, we apply SSiB4/TRIFFID-Fire to investigate fire effects

on vegetation cover, ecosystem productivity, and surface energy on monthly to annual scales in

Sect. 3. Discussions and conclusions are given in Sect. 4.

## 2 Method

### 2.1 Study region

We conduct our fire modeling study in Southern Africa (SAF; 0-35 °S, 0-50 °E) which

has a typical savanna climate and the most representative savanna fire. SAF has the largest

continuous stretch of savanna covering an area of $1.4 \times 10^3$ Mha of the land surface. The SAF

savanna has an annual burned area of 153.7 Mha yr$^{-1}$ (Giglio et al., 2018) and carbon emission of

669 Tg C yr$^{-1}$ (van der Werf et al., 2017) in climatology, contributing to about 36 % and 31 % of

the global total burned area and fire carbon emissions, respectively. Each year in the dry season,

fire leaves numerous scars on land surface and the local ecosystem has evolved with fire as an

essential contributor to its structure and function. Therefore, we select SAF as the study region to

115    quantify the fire effects at monthly to annual scales.

Over the SAF, the Equatorial Africa (0-5 °S), East Coast of SAF, and East Coast of

Madagascar Island are hot and humid throughout the year, with an annual mean temperature of

25 °C and rainfall exceeding 1200 mm yr$^{-1}$ (Fig 1a-b). From the equator to Southern Hemisphere

(SH) high latitude, the annual mean precipitation and temperature decrease while the seasonality

is enhanced (Fig 1c-d). The SAF savanna has a divergent climate during the wet season

(November-April) and dry season (May-October). During the rainy season, the daily



precipitation can reach 15 mm day$^{-1}$, resulting in significant floods in Zimbabwe, Zambia, Malawi, and Mozambique (https://reliefweb.int/report/malawi/south-east-africa-deadly-storms-and-floods-malawi-zambia-and-mozambique). The dry season for savanna includes May-

October, which is characterized by little precipitation especially for June-July-August when monthly rainfall is less than 10 mm (Fig. 1d). SAF has diverse ecosystems influenced by climate and fire. From tropical SAF to southern hemisphere high latitude, the climatology land cover ranges from the densely forested area, savanna, grassland, shrubland, and desert correspondingly (Fig. 1e). Equatorial Africa is dominated by tropical rainforests, known as the Congolese

rainforest. Most areas between 5-20 °S are dominated by C$_4$ grasses with tree fraction varying between 10-20 % with moisture conditions, referred to as the savanna biome. C$_3$ grasses are mostly distributed in the eastern part of the SAF along the Great Rift Valley and the eastern portion of the Great Escarpment. The shrub dominates the Southern African Plateau.



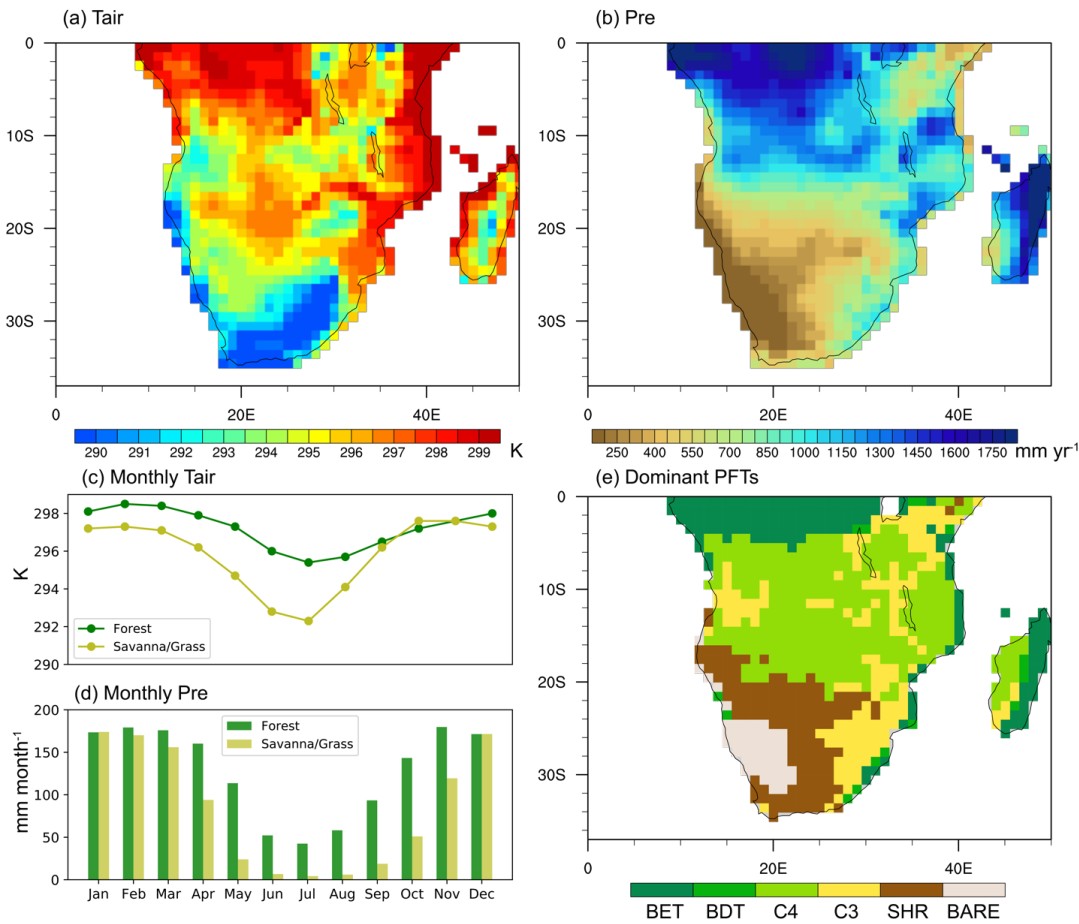

**Figure 1** (a)-(d) Climate of SAF from Sheffield et al. (2006) averaged in 2000-2013 (a) annual mean air temperature (Tair), (b) annual total precipitation (Pre), (c) monthly Tair (d) Monthly Pre, and (e) dominant PFTs for each grid simulated in SSiB4/TRIFFID-Fire (BET: broadleaf evergreen trees; BDT: broadleaf deciduous trees; C4: $C_4$ grasses; C3: $C_3$ grasses; SHR: shrub; BARE: Bare land)

## 2.2 The SSiB4/TRIFFID-Fire vegetation-fire model

SSiB4/TRIFFID-Fire consists of three components: a land surface model (Simplified Simple Biosphere Model; SSiB), a dynamic vegetation model (the Top-down Representation of Interactive Foliage and Flora Including Dynamics Model; TRIFFID), and a fire model of





intermediate complexity (Huang et al., 2020b). SSiB simulates surface radiation components,

momentum fluxes, sensible heat (SH) and latent heat (LH) fluxes, soil moisture, surface

temperature, and vegetation gross/net primary productivity (GPP/NPP) based on energy and

water balance (SSiB; Xue et al., 1991; Zhan et al., 2003). The SSiB was coupled with TRIFFID,

which describes the vegetation dynamics based on species competition for common resources

and provides an interactive component in the feedback loop of ecosystem and climate (Cox,

2001; Harper et al., 2016). The vegetation competition in TRIFFID is based on the Lotka-

Volterra (LV) equation, which has been updated in Zhang et al. (2015) to represent the

coexistence of grasses and shrubs. Liu et al. (2019a) further adjusted the large-scale disturbance

(LSD) parameter, which includes vegetation disturbances due to fires and other processes, to

allows for the coexistence between trees, $C_3$ grasses, and $C_4$ grasses. The modeled plant

functional types (PFTs) in SSiB4/TRIFFID include broadleaf evergreen trees (BET), needleleaf

evergreen trees (NET), broadleaf deciduous trees (BDT), $C_3$ grasses, $C_4$ grasses, shrubs, and

tundra. The simulated vegetation distribution in SSiB4/TRIFFID has been evaluated with

observations over Northern America (Zhang et al., 2015) and over the globe (Liu et al., 2019a).

The SSiB4/TRIFFID is further improved by incorporating a fire scheme (Li et al., 2012)

to describe fire disturbance on vegetation dynamics and carbon cycle (hereafter SSiB4/TRIFFID-

Fire). SSiB4/TRIFFID-Fire is shown to reproduce the burned area and fire emissions across the

spatial and temporal scales (Huang et al., 2020b). Specifically, it produces realistic fire peak

months and fire season length in major fire regions, including Southern Africa (SAF), South

America, Southeast Asia, and Equatorial Asia. The fractional coverage of each PFT has been

thoroughly validated with observations on the global scale. With an explicit description of the

burned area, carbon emission, and fire disturbance on vegetation, SSiB4/TRIFFID-Fire captures



fire-vegetation interactions under current climate and can be used to study fire effects on ecosystem characteristics and surface energy.

2.3 Model improvement

SSiB4/TRIFFID-Fire has been updated to improve the simulation of monthly fire regimes, vegetation productivity, and surface fluxes in SAF. A constant crop fraction from GLC2000 (Bartholome and Belward, 2005) was used to describe the agricultural land during the entire simulation period in Huang et al. (2020b). Studies show that agriculture expansion has played a role in the spatial and temporal variations of the burned area in the tropical region and should be

explicitly described (Andela et al., 2017; Lasslop and Kloster, 2017). We have introduced an annually-updated crop fraction from Land-Use Harmonization 2 (LUH2) datasets (Hurtt et al., 2006; Hurtt et al., 2011) to investigate the influence of crop fraction interannual changes on fire. Since the LUH2 has a smaller crop fraction and a different spatial distribution in tropical regions than those in GLC2000, we have calibrated the parameters of fire spread, fuel combustibility,

and carbon combustion to reproduce the observed magnitude and temporal variations of burned area and carbon emission in satellite data.

Wet season accumulated productivity proves to be one of the determinants for the burned area and carbon emission in the following fire season (Forkel et al., 2019). The vegetation productivity, however, is also influenced by phenology and fire. Our previous study found that

SSiB4/TRIFFID-Fire captured its spatial distribution and interannual variations, but the annual mean GPP was overestimated (Huang et al., 2020b). During an annual cycle, the model properly simulates the GPP magnitude in the wet season but overestimates it in the dry season, leading to an underestimation of the seasonality of ecosystem functioning in savanna and grassland. A similar conclusion can be drawn in the simulated intra-annual variations of LAI. To ensure that



the model captures the fire-vegetation-climate feedback at the monthly scale, we optimize

photosynthesis-related parameters according to the observed GPP magnitude in both wet seasons

and dry seasons as follows.

By compiling 32 years of satellite data, Li et al. (2019b) reported that moisture condition

(precipitation) was the first vital driver positively affecting monthly vegetation productivity in

non-forest areas. In SSiB4, the vegetation productivity is closely associated with the soil

moisture through the root-zone soil moisture potential factor $f(\theta)$:

$$f(\theta) = 1 - e^{-c_2[c1_1 - \ln(-ph_0 * \theta^{-b})]}, \tag{1}$$

where $c_1$, $c_2$, $ph_0$, and $b$ are PFT-dependent parameters. $f(\theta)$ represents the soil moisture ($\theta$)

effects on stomatal resistance, which influences $CO_2$ and water exchanges and can also affect

leaf turnover. $f(\theta)$ does not play a role when it is close to 1 and can largely suppress the

transpiration and vegetation productivity when it is close to 0. For $C_3$ and $C_4$ grasses, the original

$f(\theta)$ decreases sharply when soil moisture ($\theta$) is between $0.3 - 0.4$, yet it has a negligible effect

when soil moisture is higher than 0.4 (Fig. 2). In SSiB4/TRIFFID-Fire, the simulated root-zone

soil moisture is generally higher than 0.4 in the SAF dry season. Therefore, we adjusted the

coefficients $c_1$ and $c_2$ for $C_4$ grasses to reflect the effects of soil water deficit on transpiration in a

wider range of soil moisture between $0.3 - 0.6$ (Fig. 2a). $f(\theta)$ for $C_3$ grasses is also adjusted but

is designed to be less sensitive to low moisture conditions (compared to $C_4$ grasses) to make it

more adaptive in the dry area (Fig. 2b).



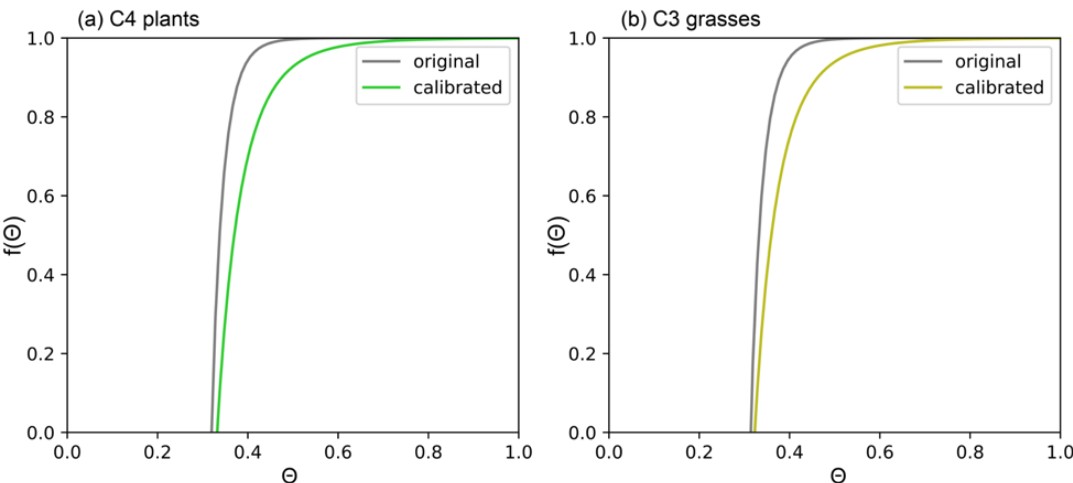

**Figure 2** $f(\theta)$ calibration for (a) $C_4$ grasses and (b) $C_3$ grasses

2.4 Experiment design

A series of offline experiments have been conducted using SSiB4/TRIFFID-Fire, including a spin-up simulation to reach quasi-equilibrium vegetation distribution and a transient run with varying climate forcings and $CO_2$ from 1948 to 2014 (Fig. 3). The spin-up simulation is conducted with 1948-1972 climatology forcing from Sheffield et al. (2006) and 1948 atmospheric $CO_2$ concentration with fire model turned on. The DGVM reaches a quasi-equilibrium status after 200 years of simulation (Fig. 3). Based on the quasi-equilibrium status, a FIREON transient run is carried out with 3-hourly meteorological forcings and yearly-updated atmospheric $CO_2$ input in SSiB4/TRIFFID-Fire with fire model turned on. The fire model requires annual agriculture, population density, and GDP information from 1948 to 2014. The spin-up and transient runs are the same as those in Huang et al. (2020b) except for the yearly-updated crop fraction. We focus on the period of 2000-2014 when the satellite observations are available for fire model validation.





To assess the fire effects at monthly and annual scales, we conduct FIREOFF simulations

branching from the FIREON simulations on January 1$^{st}$ of each year between 2000 and 2013

(Fig. 3). Each FIREOFF simulation is run for two years with the fire model switched off and all

remaining parameters and input data the same as those in FIREON. In both FIREON and

FIREOFF simulations, the vegetation distribution is allowed to respond to climate variations

while the fire disturbance is only considered in FIREON. Each 2-year simulation in FIREOFF is

regarded as an ensemble member, and there are total 14 ensemble members. The corresponding

periods in FIREON form 14 paired ensemble members. The fire impacts on vegetation properties

and surface energy balance are quantified using the differences between FIREON and FIREOFF

ensemble means.

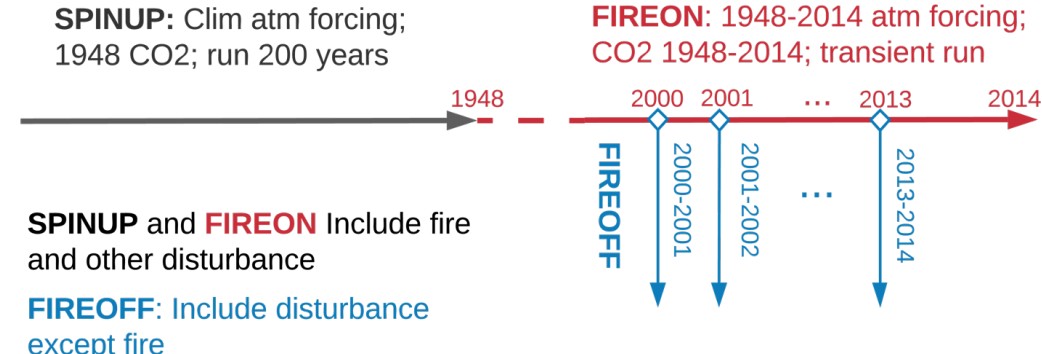


**Figure 3** Experiment Design for fire effects in SAF (0-37 ºS; 0-50 ºE)

2.5 Model input and validation data





**Table 1** Datasets used to drive SSiB4/TRIFFID-Fire and evaluate simulations

| Variables | Sources | Resolution |
|---|---|---|
| Surface air temperature | | |
| Surface pressure | | |
| Specific humidity | | |
| Wind speed | Sheffield et al. (2006) | 1°, 3-hourly |
| Downward shortwave radiation | | |
| Downward longwave radiation | | |
| Precipitation | | |
| Lightning frequency | NASA LIS/OTD v2.2 | 2.5°, 2-hourly |
| Population density | GPWv3 (CIESIN, 2005); | 0.5°, 5 yearly |
| | HYDE v3.1 (Klein Goldewijk et al., 2010) | 5', 10 yearly |
| GDP | van Vuuren et al. (2006) | 0.5°, in 2000 |
| Agriculture fraction | LUH2 (Hurtt et al., 2006; 2011) | 0.25°, yearly |
| Burned area Carbon emission | GFED4s (Randerson et al., 2012; van der Werf et al. 2017) | 0.25°, monthly |
| GPP | FLUXNET-MTE (Jung et al. 2009) | 0.5°, monthly |
| Latent heat Sensible heat | FLUXCOM (Jung et al., 2019) | 0.0833° and 0.5°, monthly |

Table 1 lists the data used for model input and evaluation. The input datasets for the fire

model and land surface model are the same as our earlier study (Huang et al., 2020b) except for

the transient crop fraction from Hurtt et al. (2006; 2011). All input datasets are interpolated to

$1.0° \times 1.0°$ spatial and 3-hourly temporal resolution to be used as model forcing.

The simulated fire variables, vegetation productivity, and surface fluxes are evaluated

against observations. The Global Fire Emission Database (GFED) is a fire dataset derived

primarily from MODIS satellite (van der Werf et al., 2006; van der Werf et al., 2010; Giglio et

al., 2013). The latest version, GFED4s, has been updated to include the contribution from small

fires below the MODIS detection limit (van der Werf et al., 2017). The burned area and carbon

emission obtained from https://www.globalfiredata.org/data.html are used to evaluate fire





simulation in SAF with a focus on monthly variations. FLUXNET Model Tree Ensemble (FLUXNET-MTE) GPP is upscaled from FLUXNET observations to the global scale using the machine learning technique MTE (Jung et al., 2011). The 1982–2011 FLUXNET-MTE GPP downloaded from https://www.bgc-jena.mpg.de/geodb/projects/Data.php has been resampled to 1.0°×1.0° to be compared with SSiB4/TRIFFID-Fire.

The FLUXCOM provides monthly gridded LH and SH estimates at 0.5° spatial resolution and monthly steps (http://www.fluxcom.org/EF-Products/). The data is derived by merging energy flux measurements from FLUXNET eddy covariance tower with remote sensing and meteorological data using machine learning techniques (Jung et al., 2019). FLUXCOM database comprises of two complementary products for surface fluxes: FLUXCOM-RS integrates the FLUXNET measurements and 2001-2015 MODIS data in machine learning techniques, while FLUXCOM-METEO estimates surface fluxes from daily meteorological data and mean seasonal cycles of satellite data. The dataset is specially designed to quantify global land-atmosphere interactions and to provide a benchmark for land surface model simulations.

## 3 Modeling fire effects in SAF

### 3.1 Model validation

We first evaluate the model simulation of burned area, carbon emission, vegetation productivity, and surface fluxes in SAF. According to GFED4s, an average of 175.6 Mha land surface is burned each year (Fig. 4a), emitting 678.9 Tg carbon into the atmosphere during 2000-2013. SSiB4/TRIFFID-Fire has captured the magnitude of annual burned area (185.8 Mha) and carbon emission (723.4 Tg C) with spatial correlation coefficients (SCC) of 0.72 and 0.78, respectively (Figs. 4b,c). Fires are mostly found in Central Africa (5 °S to 20 °S), extending from





275 the Atlantic Coast to Lake Tanganyika. The most extensive burned area is found in savanna with intermediate productivity, where the aboveground biomass and dried soil conditions in the dry season facilitate fire occurrence and spread. Fires in tropical Congolese rainforest and drylands in Namibia and South Africa are constrained by climatic conditions and fuel load, respectively. The observed burned area fraction shows some "hot spots" in Angola, Zambia, and the southern

280 part of Congo; however, the model produces more homogeneously distribution. The heterogeneity in GFED4s may come from landscape fragmentation associated with intensive agriculture, which limits the burned area by reducing fuel connectivity (Bistinas et al., 2014). Although SSiB4/TRIFFID-Fire excludes fire occurrence in agricultural land, it does not consider the effect of landscape fragmentation on fire spread and tends to underestimates the negative

285 impact of cropland on the burned area. Besides, the simulation is conducted at $1.0° \times 1.0°$ spatial resolution. The relatively coarse model resolution makes it harder to capture the spatial heterogeneity in fire distribution.

  Fire in SAF is concentrated in the dry season (Fig. 4d). Fire season in SAF is defined as June-October during which the monthly burned area contributes to more than $\frac{1}{12}$ of the annual

290 burned area (Venevsky et al., 2019). The aboveground dried fuel can be easily ignited and cause extensive fires in the dry season. The monthly burned area drops dramatically at the beginning of the rainy season and remains low until May in the next year. SAF savanna fire has a clear distinction between June-October and November-May, reflecting the contrasting climate during the rainy (non-fire) season and dry (fire) season (Fig. 1d).



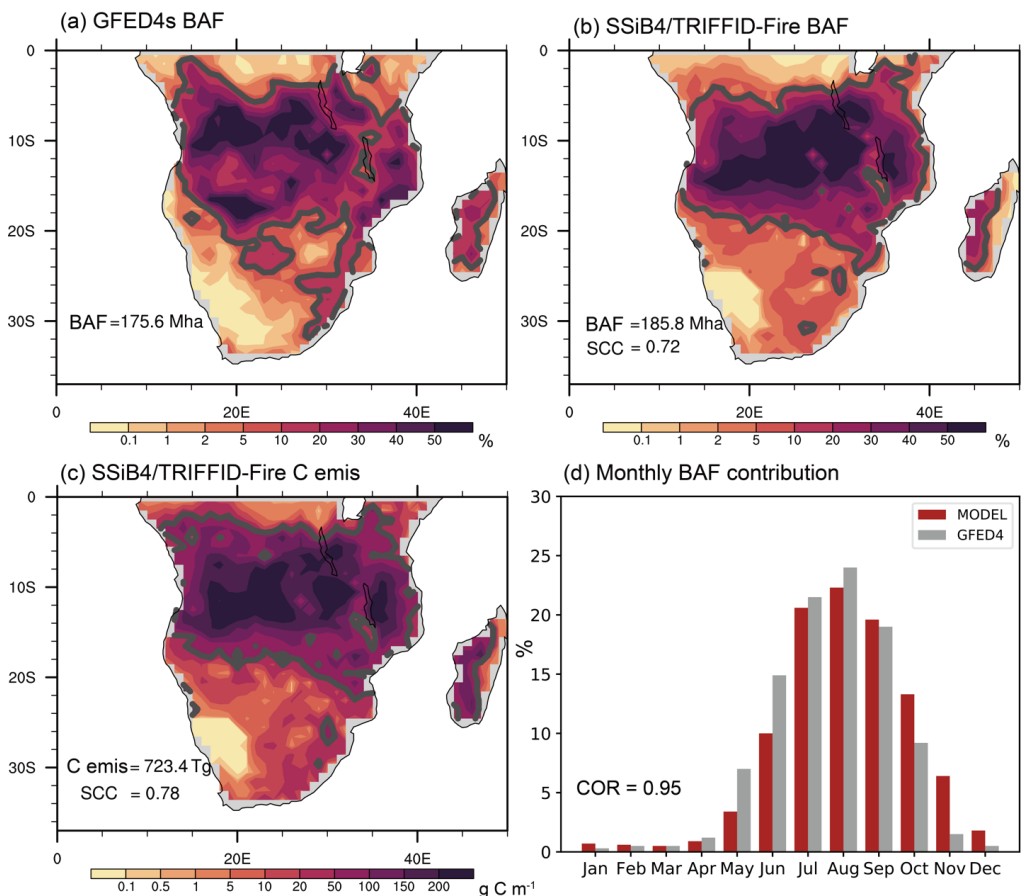

**Figure 4** (a) Annual burned area fraction (BAF; %) averaged over 2000–2013 in GFED4s, (b) same as (a) but in SSiB4/TRIFFID-Fire, (c) Annual fire carbon emission (C emis; g C m⁻¹) averaged over 2000–2013 in SSiB4/TRIFFID-Fire, and (d) Contribution of the monthly burned fraction to the annual burned fraction in model and observation (SCC: spatial correlation coefficient; COR: temporal correlation coefficient).

In the following section, we evaluate the model simulation of vegetation productivity and surface fluxes and their seasonality. The model simulation of GPP is evaluated against the FLUXNET-MTE product. The annual average GPP in 2000-2011 is 1283.4 g C m⁻² yr⁻¹, ranging from more than 2500 g C m⁻² yr⁻¹ in the tropical rainforest to less than 400 g C m⁻² yr⁻¹ in the





shrubland (Fig. 5a). The GPP magnitude and spatial distribution are simulated in SSiB4/TRIFFID-Fire (1326.3 C m$^{-2}$ yr$^{-1}$), with an SCC of 0.89 (Fig. 5b). The model has captured the monthly variations of GPP with a correlation higher than 0.7 for most grid cells ($p<0.05$; Fig. 5c), although it slightly underestimates GPP in the dry season (Fig. 5d). Observational studies

have shown that GPP in 3-6 months preceding the fire season is a vital predictor for savanna fire (Forkel et al., 2019). In return, fire can influence the vegetation productivity in the following 2-6 months after fire (Dintwe et al., 2017). SSiB4/TRIFFID-Fire uses aboveground biomass, which is related to vegetation productivity accumulation in preceding wet months, to describe the fuel constraint on fire ignition. The monthly variation of vegetation productivity in the model is in a

good consistency to that in observations (Fig. 5d), indicating that SSiB4/TRIFFID-Fire captures some critical processes in fire–vegetation-climate interactions.

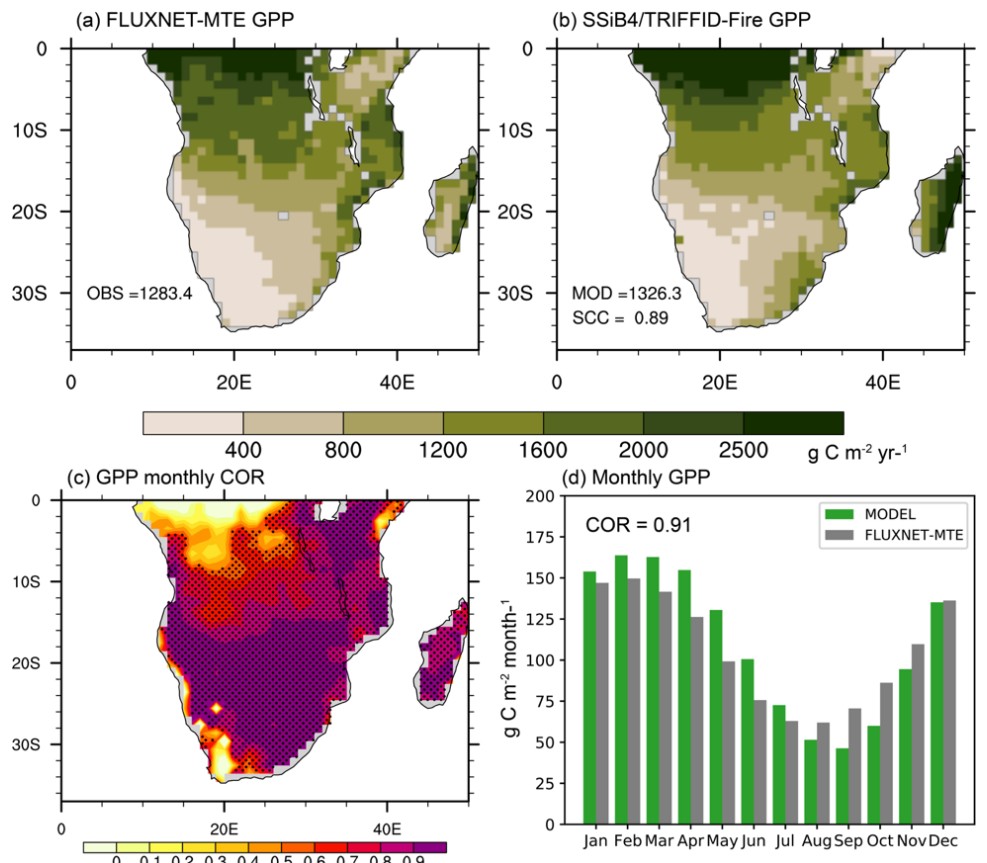

**Figure 5** Annual GPP (g C m⁻² y⁻¹) averaged over 2000–2011 in (a) FLUXNET-MTE, and (b) SSiB4/TRIFFID-Fire; (c) Point-by-point climatology monthly correlation between FLUXNET-MTE and SSiB4/TRIFFID-Fire (dots indicate the correlation is significant with p-value < 0.05); (d) Monthly GPP (g C m⁻² mon⁻¹) in model and observation

The simulated energy fluxes are compared with FLUXCOM datasets to evaluate the surface fluxes partitioning between LH and SH. The spatial distribution of LH shows a predominant latitudinal distribution with a strong N-S gradient in both FLUXCOM-METEO and model, decreasing from 100 W m⁻² in tropical Congolese forest to less than 30 W m⁻² in the Kalahari Desert (Figs. 6a,b). Regions east of Lake Tanganyika have a much smaller annual





precipitation (600 mm yr⁻¹ in Fig. 1b) and, therefore, have a smaller LH than the west part of

SAF. In contrast, SH in SAF peaks in the Kalahari Desert and gradually decreases towards the

tropical forest (Fig. 6c). SSiB4/TRIFFID-Fire captures the latitudinal distribution of SH yet

underestimates its magnitude in the desert (Fig. 6d).

We further compare the monthly variations of LH and SH in SSiB4/TRIFFID-Fire

against the observation-derived datasets, FLUXCOM-METEO and FLUXCOM-RS. There is a

high agreement in the regional average and range of LH/SH between the model and observations

(Fig. 6e-f). The LH peaks in the rainy season (December-January-February; DJF) and gradually

declines and reaches the minimum in June-July-August (JJA). SSiB4/TRIFFID-Fire tends to

underestimate LH in the wet season but accurately simulates its magnitude in the dry season. It

captures the peak of SH in September-October while slightly overestimates it in January-

February-March. Jung et al. (2019) pointed out that FLUXCOM LH estimates in Africa are

larger than other observation-based datasets. As such, the simulated LH may compensate for the

wet bias in FLUXCOM datasets. Overall, the model is shown to reproduce the annual surface

fluxes distribution and their seasonality in SAF.

The comparison with observations shows that SSiB4/TRIFFID-Fire is capable of

reproducing the annual mean and intra-annual variations of fire regimes in SAF. The seasonal

cycle of vegetation productivity and surface energy fluxes are captured, indicating the fire-

coupled vegetation model can describe some key processes in the feedback between fire,

vegetation, and surface energy. The fire-coupled DGVM is then used to provide the first model

quantification of short-term (monthly to annual scales) fire effects by assessing the differences

between FIREON and FIREOFF from June to May in the next year, including a complete fire

season (June-October) and post-fire recovery season (November to May in the next year).



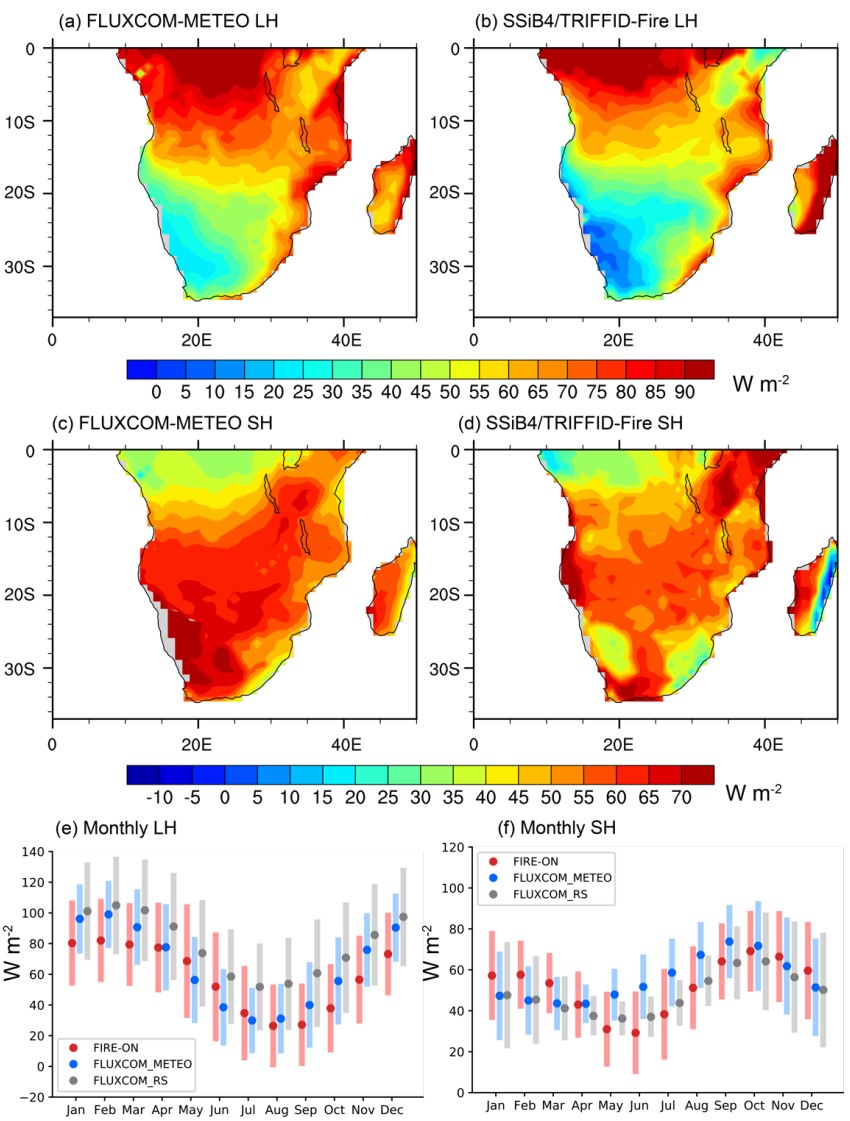

**Figure 6** Annual LH (W m⁻²) averaged over 2000–2013 in (a) FLUXCOM-METEO, and (b) SSiB4/TRIFFID-Fire; Annual SH (W m⁻²) averaged over 2000–2013 in (c) FLUXCOM-METEO, and (d) SSiB4/TRIFFID-Fire; Monthly (e) LH and (f) SH for grid within SHAF in the model, FLUXCOM-METEO, and FLUXCOM-RS. The dots in (e) and (f) denote the regional mean values, while the bars denote the LH/SH values within one standard deviation of the mean value of all grid points





## 3.2 Fire effects on vegetation

The reoccurrence of fire changes the fractional coverage of trees and short PFTs in SAF.

Figure 7a and Figure 7b show the spatial distribution of trees (BET and BDT) and grasses ($C_3$ and $C_4$ grasses) in FIREOFF. Tree PFTs cover more than 80% in equatorial Africa and the east coast of Africa, while $C_3$ and $C_4$ grasses dominate widespread areas between 5 °S and 20 °S and the eastern portion of the Great Escarpment. The reduction in tree cover ranges from 0.2 % to 0.6 % per year in Africa savanna and can exceed 1 % in the transition zone between savanna and

Congolese forest (Fig. 7c; FIREON minus FIREOFF), indicating fire is an important contributor to tropical deforestation (Hansen et al., 2013). A decrease in $C_3/C_4$ grasses fraction by 4-8 % is found in regions with an annual burned area fraction greater than 10 %, and the magnitude of change is generally proportional to the grid burned area fraction (Fig. 7d). Looking further into the grass cover change per burned area fraction, we find a larger fire impact in drier regions with

annual rainfall smaller than 600 mm $yr^{-1}$ and with GPP generally smaller than 800 g C $m^{-2}$ $yr^{-1}$ (not shown). The larger fire effects could be explained by the slower recovery after fire corresponding to the lower vegetation productivity.

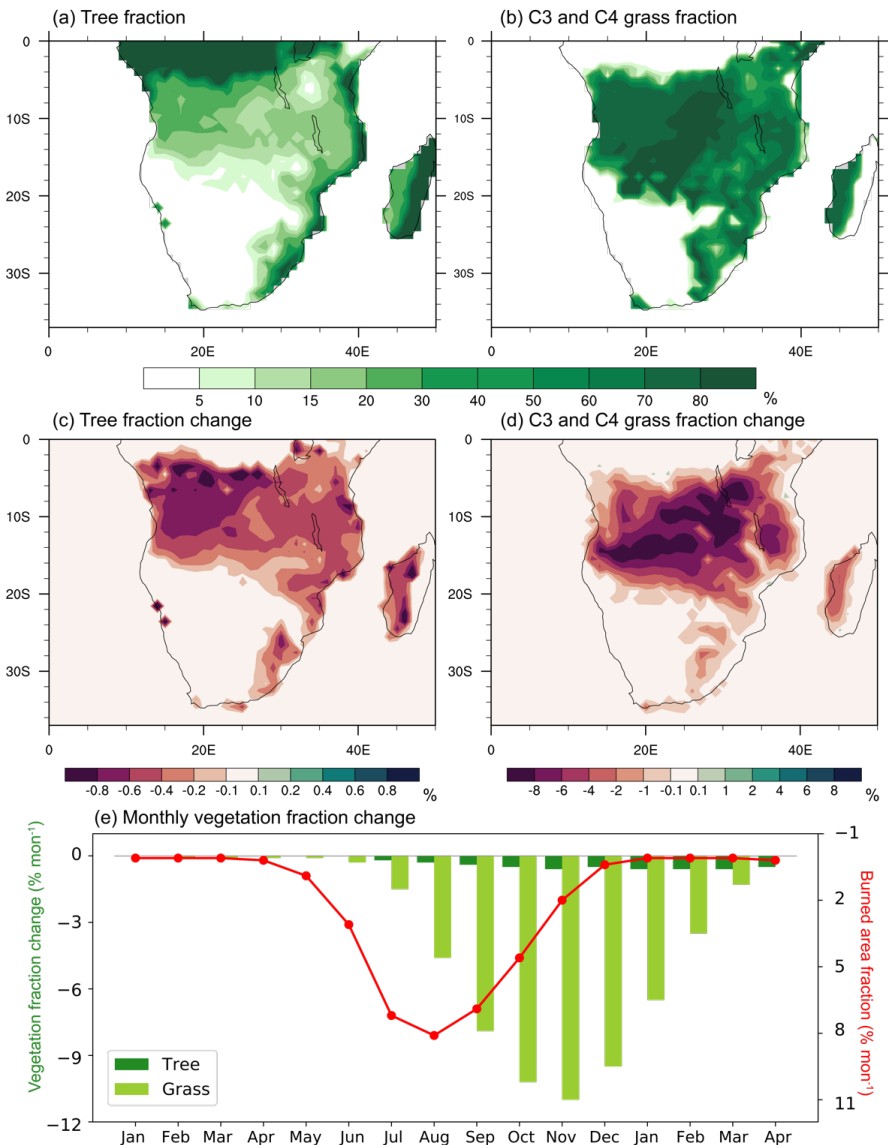

**Figure 7** 2000–2013 annual (a) tree fraction (%) and (b) C4/C3 grass fraction (%) in FIREOFF; 2000–2013 annual fire effects on (c) tree fraction and (d) grass fraction, and (e) monthly fire effects on the fractional coverage of trees and $C_4/C_3$ grasses (% mon$^{-1}$) with burned area fraction (% mon$^{-1}$, red curve) overlaid (The areal-average fire effect is calculated using grids with an annual burned area fraction > 10%). The y-axis of burned area fraction in (e) is reversed.



Figure 7e shows the monthly vegetation cover change in the fire season and the following

rainy season (FIREON minus FIREOFF). The dark green and light green bars denote changes in

tree cover and grass cover between FIREON and FIREOFF, which are linked to the monthly

burned area fraction denoted by the red line. There is little burned area fraction during the

precedent wet season from January through May, during which vegetation removal by fire is

negligible. While the monthly burned area fraction peaks in August and gradually decreases after

that, regional fire disturbance on $C_3$ and $C_4$ grasses accumulates during the entire fire season

until November. The vegetation recovery is limited during June-October when the arid

conditions produce very small vegetation productivity. In the dry season, the monthly

precipitation is generally smaller than 20 mm and GPP is 50 g C m$^{-2}$ month$^{-1}$, only $\frac{1}{3}$ of that in the

wet months (Fig. 5d). Fire effects reach the maximum in November when a decrease of 11 % is

found in the grass coverage. Vegetation recovery is accelerated in the rainy season and grass

cover reduction is gradually diminished from December to April in the following year. At the

end of the rainy season, little difference in the grass cover is found between FIREON and

FIREOFF. Observational studies support our conclusion that the vegetation in SAF savanna is

highly adaptive to fire and can mostly recover to unburned conditions within 8 days to 12 months

(Araújo et al., 2017; Dintwe et al., 2017; das Chagas and Pelicice, 2018).

Different from grass PFTs, tree cover change caused by fire barely recovers within one

growing season (Fig. 7e). This low fire adaptivity makes trees highly vulnerable to fire,

especially at the periphery of tropical forests. Our finding corroborates results from observational

studies that fire plays a key role in tropical forest loss (Cochrane et al., 1999; Hansen et al.,

2013). Meanwhile, fire-induced forest clearance facilitates the growth and spread of grasses,



allowing for the coexistence between trees and grasses in the savanna ecosystem which would otherwise be encroached by trees (Higgins et al., 2007; Furley et al., 2008).

The removal of vegetation canopy has caused a reduction in canopy area and vegetation productivity, reflected by the changes of LAI and GPP (Fig. 8a-b). For most parts of SAF

savanna, fire has caused a relative change of LAI ($\frac{\Delta LAI}{LAI_{FIREOFF}}$) by -3 % to -5 %, whose magnitude is proportional to the burned area fraction. Over regions with a burned area fraction higher than 10 %, we find a decrease of LAI by 0.10 $m^2$ $m^{-2}$ on average, accompanied by a decrease of vegetation height by 0.17 m (not shown). The fire impact on LAI accumulates in the dry season, during which the LAI consumption by fire outcompetes the recovery (Fig. 8c). The fire impact

on LAI peaks in November when a relative change by -11% is found. At the end of the recovery season, there is still a small difference by 1 % in the grid-average LAI corresponding to the long-lasting tree cover loss. A greater magnitude of relative change (-5 % to -7 %) is found in vegetation productivity (GPP and NPP) caused by fire. Overall, we find an annual average reduction of GPP and NPP by 59 g C $yr^{-1}$ and 32 g C $yr^{-1}$ in regions with a burned area fraction

higher than 10 %. The changes in vegetation cover and properties (GPP, NPP, LAI, and vegetation height) influence the radiation absorbed by the surface and the energy partitioning between LH and SH.

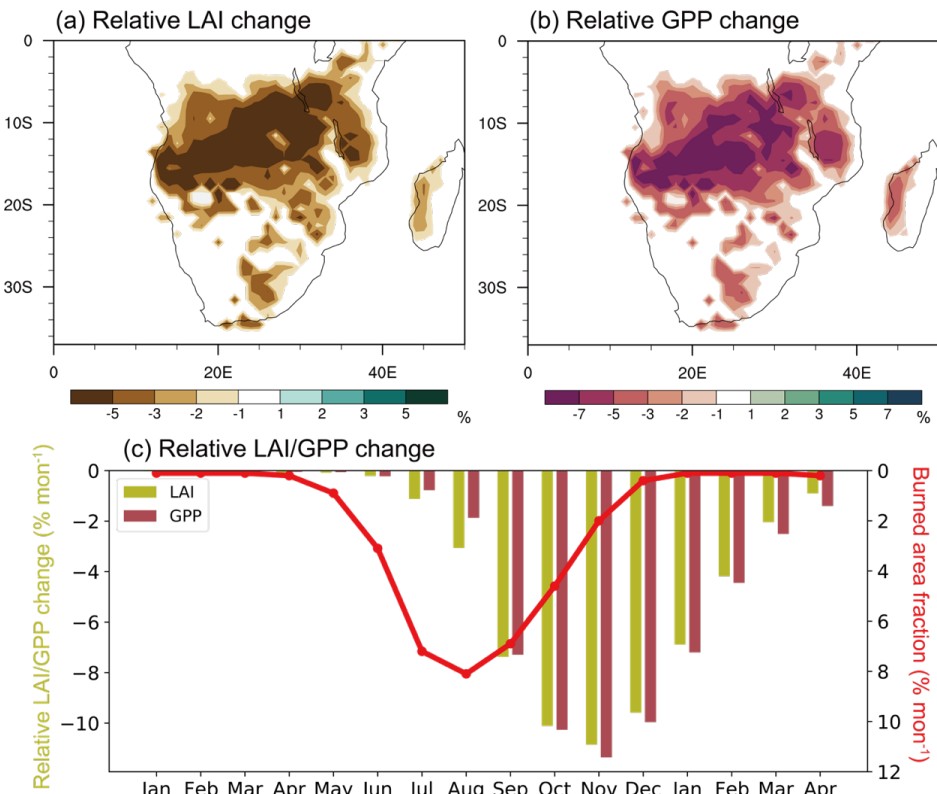

**Figure 8** 2000–2013 annual fire effects on (a) LAI (%) and (b) GPP (%), and (c) Monthly fire effects on

the LAI/GPP (% mon$^{-1}$) with burned area fraction (% mon$^{-1}$) overlaid (The areal-average fire effect is

calculated using grids with an annual burned area fraction > 10%). The y-axis of burned fraction in (c) is

reversed.

3.3 Fire effects on surface energy

425         Over the SAF, a decrease of net shortwave radiation (NSW) on surface is found by an

average of 0.60 W m$^{-2}$ (Fig. 9a). The magnitude of change ranges between -2.9 W m$^{-2}$ and 0.8 W

m$^{-2}$ (Fig. 9d), generally increasing with the annual burned area fraction. The consumption of

vegetation canopy during the fire season has caused an exposure of bare soil, which generally

has a higher reflectance than vegetation canopy. Therefore, the surface NSW is reduced between





June and November (Fig. 10a), especially in October (-1.5 W m$^{-2}$) when the arid soil has distinct

contrast in surface albedo compared to the surrounding vegetation area. The bare soil albedo is

largely decreased when the soil is moist by rain and the corresponding NSW change quickly

diminishes. The differences in NSW are mostly invisible after December.

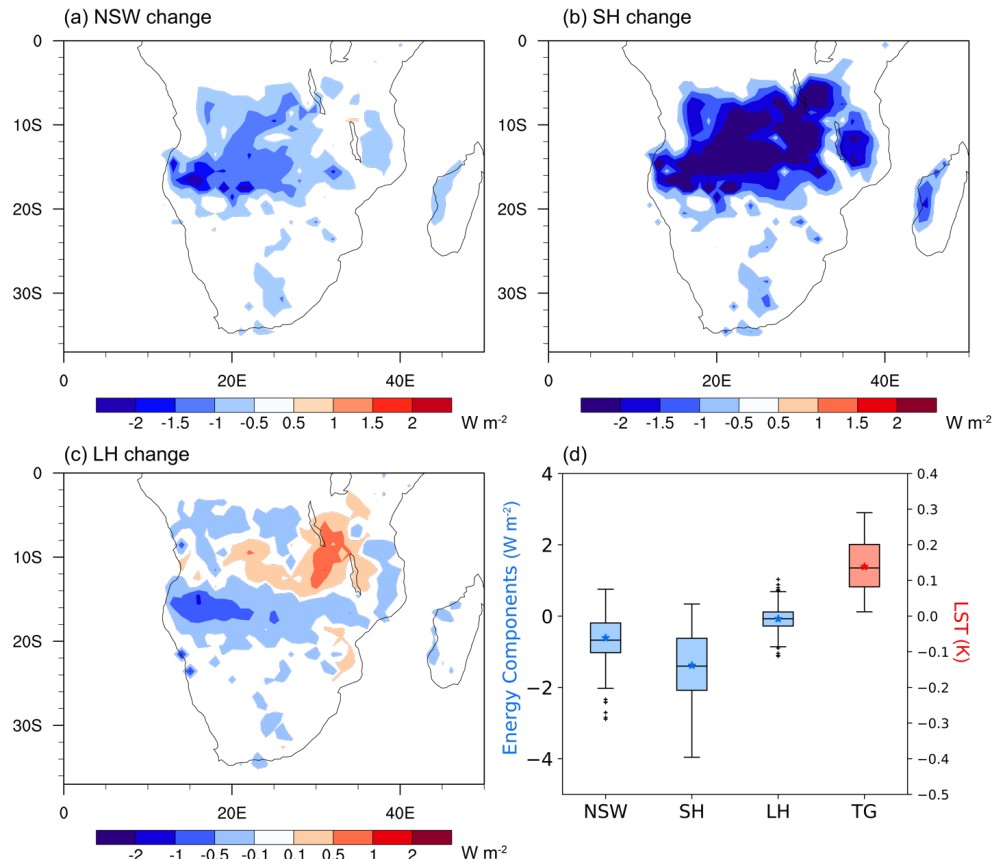

**Figure 9** 2000–2013 annual fire effects on (a) NSW (W m$^{-2}$), (b) SH (W m$^{-2}$), (c) LH (W m$^{-2}$). (d) Box

plot of annual fire effects on NSW, SH, LH, and TG (K) for grids with an annual burned fraction > 10%

within SHAF, with medians (middle bars), interquartile ranges (between 25th and 75th percentiles; boxes),

maxima/minima (whiskers) within ±1.5 × interquartile ranges, and outliers ("+"). The blue and red

asterisks (*) denote the areal-weighted mean fire effects in SHAF




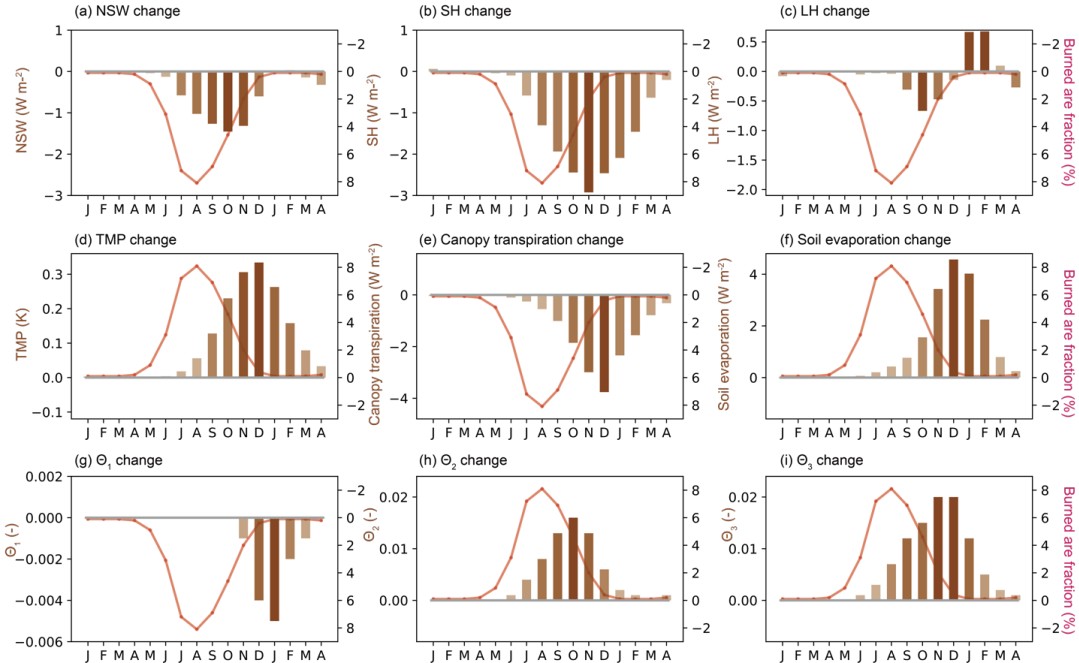

**Figure 10** 2000–2013 monthly fire effects on (a) NSW (W m$^{-2}$), (b) SH (W m$^{-2}$), (c) LH (W m$^{-2}$), (d) TMP (K), (e) canopy transpiration (W m$^{-2}$), (f) soil evaporation (W m$^{-2}$), (g) $\theta_1$, (h) $\theta_2$, and (i) $\theta_3$ with burned area fraction (%) overlaid. The y-axis of burned fraction in (a-c), (e), and (g) are reversed.


The vegetation removal after fires has reduced the grid-average vegetation height, corresponding to a decrease in surface roughness length and an increase in aerodynamic resistance. The changes of aerodynamic features influence near-surface drag force, affecting the sensible heat flux exchange between land and atmosphere (Liu et al., 2016). In this experiment,

we find a widespread reduction of SH by 0.3 – 4.0 W m$^{-2}$ with a regional average of 1.4 W m$^{-2}$ (Fig. 9b). The magnitude of change is generally proportional to the grid's annual burned area fraction (Fig. 4b). The monthly fire effect on SH peaks in November, producing a decrease by 2.9 W m$^{-2}$ in regions with an annual burned area fraction greater than 10 %. There is a small LH change (-0.1 W m$^{-2}$) at the annual scale (Fig. 9c) due to the opposite changes in canopy



transpiration by -1.9 W m$^{-2}$ and surface evaporation by 1.8 W m$^{-2}$ (discussed later). Overall, we

find a slight increase in surface temperature by 0.14 K (Fig. 9d), which peaks in December (0.33

K; Fig. 10d). Despite surface NSW is reduced after fire, the change of surface fluxes, especially

SH, seems to dominate the temperature changes in tropical savanna. Aside from this, SH and LH

are important nonradiative heating sources to the atmosphere. The decrease of SH and LH is

expected to cause an atmospheric cooling and subsidence near the surface, generating a negative

impact on convective precipitation (De Sales et al., 2018). Fire effects on precipitation are not

simulated as we use fixed precipitations from the forcings. Yet it can be anticipated that the

reduced atmospheric heating can largely suppress convective storms at the beginning of the rainy

season and even impede monsoon progression.

In fire season, a maximum LH reduction by 0.7 W m$^{-2}$ is found in October, while in the

following rainy season, the simulated LH is enhanced by 0.7 W m$^{-2}$ in February (Fig. 10c). The

change of LH is related to the opposing changes of canopy transpiration and soil evaporation.

When fire occurs, the removal of vegetation leaf area (Fig. 8a) has caused a decline in canopy

transpiration (Fig. 10e), accompanied by increases in root-zone soil moisture (Fig. 10h) and

deeper layer soil moisture (Fig. 10i) when less soil water is transported to the atmosphere.

Previous studies showed that the removal of dense plant canopy caused a smaller surface

resistance for soil evaporation (Schulze et al., 1994), especially when the soil was nearly

saturated (Dunin, 1987; Gholz and Clark, 2002). Indeed, we find an increase in soil evaporation

throughout our study period. The increase is weak during the dry season as there is not much

evaporable water (Fig. 10f), yet it is greatly enhanced after November when the soil is refilled

after rain. The elevated soil evaporation has caused a decrease in the surface layer soil moisture

(Fig. 10g), which is confined in wet months when soil is moist by rains.





4 Discussion and conclusion

4.1 Surface darkening effects due to fire

Our experiment in Sect. 3 examines the biophysical impacts of fire on surface energy
balance due to vegetation clearance. To assess how surface darkening effects influence our
previous conclusion, we conduct a sensitivity test (FIREON$_{dark}$) following the methodology in
De Sales et al. (2018): In FIREON$_{dark}$, surface albedo is reduced to 0.1 for 60 days, after which
albedo is returned to the unburned condition to mimic the removal of ash and charcoal by wind
and precipitation. The value of the darkening period is taken from Saha et al. (2019) which
showed that brightening occurs after 60 days on average in SAF.

The inclusion of soil darkening effects does not affect the simulation of the annual burned
area, carbon emission, and GPP, as we find the relative differences less than 0.002% and SCC
higher than 0.99 between FIREON$_{dark}$ and FIREON. The difference of LAI and GPP from
FIREON$_{dark}$ minus FIREOFF are -0.10 m$^2$ m$^{-2}$ and -61 g C yr$^{-1}$, respectively, with the maximum
difference occurring in November. The annual mean fire effects on vegetation productivity and
their monthly variations are highly consistent with those we find in FIREON minus FIREOFF.
When soil darkening is considered, we find an annual increase of NSW by 0.07 W m$^{-2}$ in SAF,
which is opposite to our previous findings that fire has decreased NSW by 0.60 W m$^{-2}$. Despite
the opposite change in NSW associated with the darkened surface, the responses in SH and LH
are consistent with/without darkening effects. We find a decrease in SH and LH by 1.0 W m$^{-2}$
and 0.1 W m$^{-2}$ between FIREON$_{dark}$ and FIREOFF, respectively, and the corresponding change
for canopy transpiration and soil evaporation is -1.9 W m$^{-2}$ and 1.8 W m$^{-2}$. An increase in TMP
by 0.17 K is simulated when the surface darkening effect is included, slightly higher than the





difference between FIREON and FIREOFF (0.14 K). Although NSW changes are opposite with
and without soil darkening effects, the similar magnitude of change in SH and LH indicates that
surface flux changes are dominated by aerodynamic/canopy resistance rather than surface
radiation in the tropics, in accordance with our previous finding in a land degradation experiment
(Huang et al., 2020a).

We acknowledge that uncertainties may be induced as we assigned a 60-day recovery
period and an albedo of 0.1 to mimic soil darkening effects for all pixels regardless of the
background climate, vegetation type, soil properties, and the season fire occurs. All these factors
may play a role in the immediate albedo anomalies after ash deposition, the amount of
brightening, and the evolution of radiative forcing after fire (Dintwe et al., 2017; Saha et al.,
2017). Due to the limited number of observations and the variety in post-fire albedo anomalies,
we are currently unable to constrain the uncertainty in the description of the surface darkening
effects. Therefore, the purpose of sensitivity tests is to investigate how surface darkening effects
affect our conclusions in Sect. 3, rather than to provide a quantitative estimate on the
uncertainties range induced by surface darkening.

4.2 Limitation and uncertainty

This is an offline study without atmospheric feedback. As such the fire effect is not fully
assessed. Previous studies show that atmospheric feedback may influence the SH and LH
changes by altering the temperature and moisture gradient between land and atmosphere (Huang
et al., 2020a), which cannot be represented in offline simulations as we used fixed atmospheric
conditions from the forcing. However, we expect our findings to be valid in coupled-model
simulations as surface fluxes changes by fire are shown to be dominated by resistance changes.
Besides, De Sales et al. (2018) showed that fire decreased the atmospheric convective instability,





which subsequently suppressed precipitation in the following rainy season. The precipitation

changes may exert negative feedback on evapotranspiration and vegetation recovery. A full

description of the feedback between fire, vegetation, and climate needs to be accomplished next

in atmospheric models coupled with fire-vegetation models to further understand the interactions.

There are also some uncertainties in this study due to the model-dependent representation

of fire effects on vegetation. The fire impact in SSiB4/TRIFFID-Fire is represented in a

relatively simple way using the PFT-specific combustion completeness and mortality factors.

This study has made extensive validation of fire regimes and vegetation productivity to make

sure that the model captured key processes in fire-vegetation interactions at monthly to annual

scales. We notice, however, that some models (for example, MCFIRE and SPITFIRE) have more

complex process-based treatments of post-fire effects in which tree damage is determined by fire

intensity, residence time, and tree canopy height (Lenihan et al., 1998; Thonicke et al., 2010).

These models are supposed to capture the varying fire effects for trees of different heights, as

observational studies reported greater damage for younger and smaller trees whose crown could

be completely scorched by fire (Higgins et al., 2000; Sankaran et al., 2008). However, it is still

unclear if these sophisticated treatments improve the simulation of fire regimes (Hantson et al.,

2020) and adequately produce vegetation responses to fire at different time scales. To evaluate

the robustness of model results and address the uncertainties in the simulated fire effects, we

argue that, more simulations should be conducted using different land surface models, DGVM,

and fire models to quantify the short-term fire effects on vegetation dynamics and surface energy

budget.



### 4.3 Conclusions


Fire modifies vegetation dynamics and surface properties. These biogeophysical effects influence the energy fluxes exchanges, hydrology cycle, and regional and global climate. A property quantification of the short-term fire effects is critical to understand the role of fire in Earth's ecosystem and climate and to provide proper information for societal mitigation activity.

This study applied the SSiB4/TRIFFID-Fire to investigate the monthly to annual scale fire impact in Southern Africa, where fire acts as an essential determinant to the structure and functioning of the local ecosystem. The model is shown to reproduce the fire regimes, vegetation productivity, and surface fluxes compared to observation-derived datasets. A sensitivity test is also conducted to assess the possible soil darkening effects on the model simulated fire impact.

Fire has caused an annual reduction in grass cover by 4-8% for most areas in SAF savanna. The largest reduction is mostly found at the beginning of the rainy season (November), which quickly diminishes before the next fire season. The reduction of tree cover is concentrated at the transition zone between tropical forests and savannas and is irreversible within one season. The low fire adaptivity of tree PFTs makes it highly vulnerable to fire and can cause large-scale

deforestation in extreme years. The canopy removal has caused an annual reduction in LAI and GPP by 3-5 % by 5-7 %, respectively. The largest productivity change is found in November when both LAI and GPP are reduced by 11 %.

The bare soil exposure after fire has caused an increase in albedo and thus a decrease of net shortwave radiation absorbed by the surface. Sensible heat is decreased by 1.4 W m$^{-2}$ due to

an increase in aerodynamic resistance. Canopy transpiration has dropped as well, which, however, is compensated by the increase in soil evaporation, producing a small annual effect on latent heat (0.1 W m$^{-2}$). The fire impact on vegetation, surface fluxes, and soil moisture are



highly consistent in simulations with/without the descriptions of charcoal deposition. Yet the incorporation of surface darkening effects has enhanced the net shortwave absorption, therefore

elevating surface warming effects due to fire.

**Code availability**: The source code of fire model is archived https://zenodo.org/record/4670922#.YG4lJ2T0lqs (DOI: 10.5281/zenodo.4670922)

**Author contributions:** HH conducted the simulation under the suggestions from YL and drafted the text and made the figures. All authors (HH, YX, YL, FL, and GO) have contributed to the analysis methods and to the text.

**Competing interests:** The authors declare that they have no conflict of interest.


**Acknowledgments:** This work is supported by NSF Grant AGS-1419526 and AGS-1849654. The authors acknowledge Cheyenne (doi:10.5065/D6RX99HX) provided by NCAR CISL, for providing HPC resources. FL acknowledges support from the National Key R&D Program of China (2017YFA0604302 and 2017YFA0604804) and National Natural Science Foundation of
China (41875137).



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
