# Peer review of "Modeling the short-term fire effects on vegetation dynamics and surface energy in"

_Geoscientific Model Development, 2021_

## Author Comment (AC1)

**Reviewer #1:**

This study investigates seasonal patterns of fire using the SSiB4/TRIFFID-Fire model. Thereby analyses focus on variables representing the energy balance such as latent and sensible heat. While the topic is generally interesting, I have several comments.

We thank the reviewer for providing constructive comments to improve our analyses. We have made more clarifications and explanations based on reviewer's comments. Below we provide a point-by-point response to the reviewer's comments. In the following paragraphs, the reviewer's comments are in black font and our responses are in blue.

The study is motivated by the lack of short-term studies of fire impact and analyses at monthly temporal resolution. For example l 53 "the short-term fire effects at monthly to annual scales have not been quantified in current fire-vegetation models". This is misleading because many models do include temporal aspect in fire models and monthly variability has been studied, even by the authors of this study: Huang et al 2020 GMD Fig 6,9; Fang et al 2010 Atm Chem and Phys, Fig 4. This statement and similar statements in the manuscript should be revised to clarify differences and novel aspects (eg energy balance) and that seasonal aspects are represented in other models as well.

R: This study focuses on the abrupt changes of vegetation status and surface energy before and after a fire season in the real world. This type of fire effect is also referred to as the "instantaneous effect" in Lasslop et al. (2020) but there is no common terminology for it. Many studies using remote sensing data have shown the important role of this type of fire effect  (Liu et al., 2018; Saha et al., 2019; Dintwe et al., 2017). Based on our knowledge, it has not been investigated using fire models. As such, we use the term "short-term fire effect" to distinguish this effect from "long-term fire effect," which compares the status with/without fire occurrence for hundreds of years.

In the "long-term fire effects" studies (Bond and Midgley, 2012; Seo and Kim, 2019; Li et al., 2014; Yue et al., 2015; Lasslop et al., 2020; Li et al., 2017; Huang et al., 2020; Li and Lawrence, 2017), simulations are conducted with fire model turned on and off for hundreds of years. As a result, the differences between reference simulations and control simulations include not only abrupt changes of vegetation after a fire season but also the changes of ecosystem equilibrium status accumulated from hundreds of years' fire effects (for example, the African savanna in the fire-off experiments would become forest). In the "short-term fire effect" study, we turned off the fire model for only 2 years to look at how vegetation is removed by fire in the fire season and the subsequent recovery in the following rainy season. Therefore, there is a very different focus from studying temporal aspects, including monthly variability, of the "long-term fire effects."

We recognize that the short-term fire effects are not clearly defined in the previous manuscript. We now add more clarification in Lines 59 to 62 to distinguish it from the long-term fire effects.

Fig 7 and others suggest that there is a huge mismatch between burnt area and considered state variables. The peek of state variables occurs around 3 month after the peak of burnt area. One would expect a more or less immediate effect of fire on vegetation cover as fire removes biomass within a few days. It seems that vegetation is hardly vulnerable to direct fire impacts and mortality. How can this be explained? I think that a more detailed model description (particularly of the fire model) should be added to be able to understand the behavior. Further this should be in the explained discussion section. Given these delayed responses, I'm not convinced that the model can actually simulate fire seasonality and its impacts on the energy balance adequately.

R: We agree that the fire has an immediate effect on vegetation cover and surface conditions where it occurs. However, it does not conflict with our model results. In the fire model with a spatial resolution of 1 degree per grid (about 100 x 100 km), a grid can have fire events occurring and vegetation cover decreasing in parts of the grid while vegetation growing in other parts. The overall vegetation cover change for a grid is the combined result due to fire removal and growth expansion which depends on meteorological conditions and species.

In Southern Africa savanna, water supply is one of the major factors controlling vegetation growth and also contributing to fire occurrences. During June-November, **vegetation loss due to fire > vegetation recovery**, and therefore the grid-average grass and tree cover keep decreasing (Fig. 7e) with more bare soil exposed. It is not until December that enriched rain facilitates vegetation recovery and grass cover gradually increases from previous months. The above explanation has been added to Lines 402-406

The results section contains several sentences and points that are more appropriate for the discussion, eg how the results compare to previous studies. The discussion, in turn, contains results of sensitivity analysis (FIREONdark simulations) that has not been introduced previously. I suggest to describe these experiments in the methods section and report results in the results section and therefor focus on interpretation and discussion of the results in the discussion section.

R: Per reviewer's comments, we introduce the sensitivity experiments (FIREON$_{dark}$) in the method sections and present the results in the results section (Section 3.4 Darkening effects due to ash deposition). Yet the comparison between model and observational studies are presented in multiple paragraphs after introducing the simulated results of each energy component. We feel it is better to validate model results after discussing each variable rather than merging them in the discussion section.

Generally the manuscript provides many results in detail, however, I think that the study could strongly benefit from a specific key questions or hypotheses. Such questions could deal with the expected fire impacts on the energy balance or the expected outcome of surface darkening in FIREONdark simulations.

R: Thank you for the suggestion. We agree it could be beneficial to clearly state our questions. The key questions we focus on are:

(1) How does fire impact on surface energy balance in Southern Africa through its disturbance on vegetation? And how does fire impact evolve during fire season and in the subsequent rainy season?

(2) How do the surface darkening effects influence our conclusions in (1)?

We have rephrased the above questions in the abstract (Lines 16-19) and the last paragraph of the introduction (Lines 98-101).

The manuscript also requires careful language editing. Some points are listed below.

l 23 "vegetation clearance" I suggest to reword here and elsewhere, a 1% reduction tree cover and 4-8% reduction is (at leas for me) not a vegetation clearance; there are still 99% of trees left.

R: We have done language editing for the whole text. The word "vegetation clearance" is changed to "vegetation removal" here and in the rest of the manuscript.

l 29 I suggest to add some conclusions to the abstract.

R: We have added the following discussions to the end of the abstract (Line 31 to Line 34):

"Our results suggest that fire effects in grass-dominant areas diminish within one year due to the high resilience of grasses after fire. Yet fire effects in tree-dominant areas, especially the periphery of tropical forests, are irreversible within one growing season and can cause large-scale deforestation if accumulated for hundreds of years.

l 31 reword "Earth's ecosystem" to ecosystems, I would argue that there is not one single Earth ecosystem but multiple ecosystems.

R: Thank you. It has been changed to "Earth's ecosystems"

l 35 in the boundary layer.

R: Revised as suggested.

l 49 "These STUDIEAS" reword, this suggest that all studies in previous sentence are "world without fire" studies.

R: It has been reworded as:

"Previous modeling studies on fire effects mostly focus on the "long-term fire effects" where the simulations with fire are compared with reference simulations representing "a world without fire" (Lasslop et al., 2020)."

l 63 reword/delete "we start with", does that relate to the presented study or are further studies planned?

R: It is related to the present study. The words have been deleted.

l 78 "These variations can be explained by ...".

R: The sentence has been modified.

l 92 I suggest to reword and delete "makes first attempt" and again, this is not the first attempt to simulate temporal variability of fire but to conduct the specific analyses related to the energy balance.

R: Our study of "short-term fire effect" is different from the study of long-term fire effects at the seasonal scale which compares FIREON with a scenario representing "a world without fire" (Huang et al. 2020). In the long-term fire effects, the differences between FIREON and FIREOFF simulations are caused by both fire-induced abrupt change in one year and changes of ecosystem (equilibrium vegetation status) after hundreds of years' fire effects. In the study of short-term fire effects, we only focus on the abrupt changes of vegetation and subsequently surface flux changes before and after a fire. To our knowledge, no fire models have been used to investigate this short-term fire effect.

l. 104 as mentioned, the manuscript could benefit some specific questions or hypotheses.

R: We agree. Please see our response to your major comments.

l. 108 not entire Southern Africa has typical savanna climate, for example the Fynbos is a winter rainfall region and the western parts are deserts.

R: The sentence has been revised to: "We conduct our fire modeling study in Southern Africa (SAF; 0-35 °S, 0-50 °E), where most areas have a typical savanna climate and high incidence of fires."

l. 108 what does "most representative savanna fire" mean?

R: It has been changed to "high incidence of fires". Savanna has sufficient fuel load and a divergent climate during the wet and dry seasons, which is favorable for large fire occurrence and is a typical study region for fire.

l. 111 delete "in climatology"?

R: Amended as suggested.

l. 113 "and local ecosystems have evolved".

R: Amended as suggested

l. 116 "Over SAF, ...".

R: Point taken.

l. 124 dry season mentioned previously.

R: The sentence has been reworded to avoid repeat wording.

l. 130 Shrubs dominate the...".

R: Corrected.

l. 157 tundra is not a PFT but a vegetation/biome type.

R: Type 6 has been corrected to "tundra shrubs". Our previous work separated tundra shrubs from the original single shrub category to better reflect the arctic biomes (Zhang et al., 2015; Liu et al., 2019a).

l. 170 "has been updated" I understood that this has been updated in previous study and suggest to clarify.

R: Sorry for the confusion. The model has been updated and calibrated in this study to improve fire simulations in SAF. It has been rewritten as: "This study updates the SSiB4/TRIFFID-Fire to improve the simulation of monthly fire regimes, vegetation productivity, and surface fluxes in SAF."

l. 194 what is a "vital driver"?

R: The "first vital driver" mentioned in Li et al. (2019b) refers to the most important factor that influences vegetation productivity. Li et al. (2019b) applied a multivariate regression model to investigate the relative contribution of different climate drivers to the photosynthetic activities of different vegetation types. They concluded that precipitation is the most important driver to the photosynthesis processes of woody savannas, savannas, and grasslands.

l. 176, 204 it seems that the inclusion of land use and modification of the soil moisture function are the only model developments presented in the study. Apart from that, the simulations were done using an existing model version. The model development aspect of the manuscript seems to be minor.

R: The previous version of SSiB4/TRIFFID-Fire has been developed to simulate global long-term fire-vegetation interactions. In this work, we have our focus on Southern Africa and short-term fire effects, which requires necessary model development to simulate the regional fire spatial distribution and monthly variations. We have made the following major upgrades.

Cropland fires generally have a small extent and are well managed compared to natural fires. Cropland fire is excluded in Huang et al. (2020) using the constant crop fraction data for the year 2000 from the GLC2000 dataset. In this work, we update the treatment of cropland with the annually updated crop fraction from LUH2.

The overestimation of GPP has been a challenge for dynamic vegetation models for a long time (Piao et al., 2013). The previous version of SSiB4/TRIFFID-Fire tends to overestimate GPP in the dry season and does not well capture the seasonal cycle GPP in SAF (left figure of Fig. R1). By evaluating the major controlling factors over SAF, we found the root-zone soil moisture potential factor $f(\theta)$ controls the photosynthesis process via stomate activities. Besides, $f(\theta)$ is also an important factor influencing vegetation regrowth after fire disturbance which is directly linked to vegetation productivity. Therefore, we have made great efforts to adjust $f(\theta)$ by considering the photosynthesis process of each PFT, PFT competition, fire-vegetation interactions, as well as surface hydraulic processes. After the adjustment, GPP overestimation has been largely reduced, and the seasonal cycle is well captured (Fig. R1), as well as improvements in other fields related to $f(\theta)$. Moreover, moisture constraint on vegetation productivity has been commonly parameterized in most dynamic vegetation models but has large uncertainties. We believe our finding is process-based and worth sharing with the model development community.

Per your concern, we have elaborated on the model development aspect in the revised manuscript (Lines 185-188; Lines 196-199; Lines 213-218).

[Figure]

Fig. R1 Monthly GPP (g C m-2 mon-1) in model and observation (a) before calibration and (b) after calibration

l. 204, 206 how was the adjustment done? Please clarify.

R: In Eq.1, c1 represents the wilting point at which stomates close completely, and c2 is a slope factor controlling how sensitive the vegetation responds in a dryer condition. For both C3 and C4, we have decreased c2 to reduce the slope so that the vegetation responds to drought at relatively wetter conditions (see Fig. 2 original and calibrated lines). The c1 for C3 is set to be smaller than that for C4 so that its stomates closely at a drier

condition and it is more sustainable to drought than C4. The adjustment is clarified in Lines 213 to 218

l. 215, 216 I suggest to add CO2 values.

R: We use 310.325 ppm for 1948 $CO_2$ in the spin-up simulation and a $CO_2$ time series from 310.325 ppm to 398.99 ppm in the transient run from 1948 to 2014. The range is added in the revised version

l. 225 not clear to me why fire was suppressed only for 2 year periods, why not for a longer period? Please clarify and explain why this experimental setup/modeling protocol has been chosen and developed.

R: As we respond to your major comment, the purpose of this study is to assess the fire-induced abrupt changes of vegetation and surface energy in a fire year. If we turn off fire for a longer period, the equilibrium vegetation states have been changed and will be more likely an experiment for long-term fire effects. We designed the experiment by mimicking satellite observational studies which compare a burned grid with the surrounding unburned grids in a year after the fire occurrence (Saha et al., 2019; Saha et al., 2016; Liu et al., 2019b; Gatebe et al., 2014). Therefore, our FIREOFF simulation has fire turned off only for 2 years and we focus on the fire effects from June to May in next year, including a complete fire season (June-October) and the subsequent recovery season (November-May). Fire before June is very limited and the effects are negligible, so we simply take the FIREOFF starting date as Jan 1st each year with initial conditions from FIREON.

l 359 specify "short PFTs".

R: The short PFTs refer to C3 and C4 grasses. It has been clarified.

l 365 I would not denote fire as an important "contributor to deforestation" given that it only removes 0.2 to 0.6% per year (per fire?). At what % does vegetation recover until the next fire? Does such a fire regime have long term impacts on vegetation or is vegetation in equilibrium because regrowth compensates fire removal?

R: The 0.2 to 0.6% is the net loss after one fire season and one recovery season, which equals to the fire-induced vegetation loss minus vegetation recovery averaged during June-May. In other words, the tree cover will grow by 0.2-0.6% when fire is turned off in Southern African savanna. Since fire occurs every year in South African savanna, the 0.2 to 0.6% (and 1% in transition areas) in one year will accumulate to a remarkable amount of vegetation loss, producing deforestation. For example, if fire model is turned off for hundreds of years, savannas will become forests, which is the long-term fire effects shown in other fire modeling studies.

l 400-402 does that relate to model results or reality?

R: The model simulation results shown here are supported by observational studies that fire-caused tree fraction loss results in vacant space which can be occupied by fire-resistant grass PFTs, allowing for the coexistence between trees and grasses in the savanna ecosystems.

l 409 "LAI consumption by fire" I would argue that fire does not directly consume LAI, rather fire removes biomass and causes vegetation mortality which in turn implies changes of LAI. I suggest to reword.

R: The "consumption" has been changed to "reduction" accordingly.

l 465 "In the fire season".

R: Done.

l 559 "highly vulnerable to fire ... large-scale deforestation" as mentioned, 0.2-0.6% don't suggest that trees are highly vulnerable.

R: Please see our response to your previous comment.

l 571 I suggest to add some conclusions and take home messages.

R: We have highlighted our conclusions in the last paragraph of the manuscript (Lines 586 to 592):

"Our results provide quantitative assessments of the regional fire effects over Southern Africa and highlight their distinct characteristics on trees and grasses. For grass-dominant areas where fires consume more than 10% vegetation cover each year, fire effects on surface energy generally diminish within the rainy season, reflecting the high resilience of grasses to fire. In contrast, while forest fires have smaller burned area, their effects are irreversible within one growing season and can cause large-scale deforestation at forests boundaries if accumulated for hundreds of years."

Reference:

Bond, W. J. and Midgley, G. F.: Carbon dioxide and the uneasy interactions of trees and savannah grasses, Philos T R Soc B, 367, 601-612, 10.1098/rstb.2011.0182, 2012.

Dintwe, K., Okin, G. S., and Xue, Y. K.: Fire-induced albedo change and surface radiative forcing in sub-Saharan Africa savanna ecosystems: Implications for the energy balance, J Geophys Res-Atmos, 122, 6186-6201, 10.1002/2016jd026318, 2017.

Gatebe, C. K., Ichoku, C. M., Poudyal, R., Roman, M. O., and Wilcox, E.: Surface albedo darkening from wildfires in northern sub-Saharan Africa, Environ Res Lett, 9, 2014.

Huang, H., Xue, Y., Li, F., and Liu, Y.: Modeling long-term fire impact on ecosystem characteristics and surface energy using a process-based vegetation–fire model SSiB4/TRIFFID-Fire v1.0, Geosci. Model Dev., 13, 6029-6050, 10.5194/gmd-13-6029-2020, 2020.

Lasslop, G., Hantson, S., Harrison, S. P., Bachelet, D., Burton, C., Forkel, M., Forrest, M., Li, F., Melton, J. R., Yue, C., Archibald, S., Scheiter, S., Arneth, A., Hickler, T., and Sitch, S.: Global ecosystems and fire: Multi-model assessment of fire-induced tree-cover and carbon storage reduction, Global Change Biol, n/a, 10.1111/gcb.15160, 2020.

Li, F. and Lawrence, D. M.: Role of Fire in the Global Land Water Budget during the Twentieth Century due to Changing Ecosystems, J Climate, 30, 1893-1908, 10.1175/Jcli-D-16-0460.1, 2017.

Li, F., Bond-Lamberty, B., and Levis, S.: Quantifying the role of fire in the Earth system - Part 2: Impact on the net carbon balance of global terrestrial ecosystems for the 20th century, Biogeosciences, 11, 1345-1360, 10.5194/bg-11-1345-2014, 2014.

Li, F., Lawrence, D. M., and Bond-Lamberty, B.: Impact of fire on global land surface air temperature and energy budget for the 20th century due to changes within ecosystems (vol 12, 044014, 2017), Environ Res Lett, 12, 10.1088/1748-9326/aa727f, 2017.

Liu, Y., Xue, Y., MacDonald, G., Cox, P., and Zhang, Z.: Global vegetation variability and its response to elevated CO2, global warming, and climate variability – a study using the offline SSiB4/TRIFFID model and satellite data, Earth Syst. Dynam., 10, 9-29, 10.5194/esd-10-9-2019, 2019a.

Liu, Z. H., Ballantyne, A. P., and Cooper, L. A.: Increases in Land Surface Temperature in Response to Fire in Siberian Boreal Forests and Their Attribution to Biophysical Processes, Geophys Res Lett, 45, 6485-6494, 2018.

Liu, Z. H., Ballantyne, A. P., and Cooper, L. A.: Biophysical feedback of global forest fires on surface temperature, Nat Commun, 10, 2019b.

Piao, S., Sitch, S., Ciais, P., Friedlingstein, P., Peylin, P., Wang, X., Ahlström, A., Anav, A., Canadell, J. G., and Cong, N.: Evaluation of terrestrial carbon cycle models for their response to climate variability and to CO2 trends, Global Change Biol, 19, 2117-2132, 2013.

Saha, M. V., D'Odorico, P., and Scanlon, T. M.: Kalahari Wildfires Drive Continental Post-Fire Brightening in Sub-Saharan Africa, Remote Sens-Basel, 11, 10.3390/rs11091090, 2019.

Saha, M. V., Scanlon, T. M., and D'Odorico, P.: Suppression of rainfall by fires in African drylands, Geophys Res Lett, 43, 8527-8533, 10.1002/2016gl069855, 2016.

Seo, H. and Kim, Y.: Interactive impacts of fire and vegetation dynamics on global carbon and water budget using Community Land Model version 4.5, Geosci Model Dev, 12, 457-472, 10.5194/gmd-12-457-2019, 2019.

Yue, C., Ciais, P., Cadule, P., Thonicke, K., and van Leeuwen, T. T.: Modelling the role of fires in the terrestrial carbon balance by incorporating SPITFIRE into the global vegetation model ORCHIDEE - Part 2: Carbon emissions and the role of fires in the global carbon balance, Geosci Model Dev, 8, 1321-1338, 10.5194/gmd-8-1321-2015, 2015.

Zhang, Z., Xue, Y., MacDonald, G., Cox, P. M., and Collatz, G. J.: Investigation of North American vegetation variability under recent climate: A study using the SSiB4/TRIFFID biophysical/dynamic vegetation model, Journal of Geophysical Research: Atmospheres, 120, 1300-1321, https://doi.org/10.1002/2014JD021963, 2015.

---

## Author Comment (AC2)

Reviewer #2

The authors modified the DGVM model in order to evaluate carbon emissions, vegetation and flux changes by fire in south Africa.

The simulation design, and the results are very clear and understandable, and the manuscript is basically well written. I little bit concern about description of the improved model.

We greatly appreciate the reviewer's positive and constructive comments to improve the paper writing. Below we provide a point-by-point response to reviewer's comments. In the following paragraphs, the reviewer's comments are in black font and our responses are in blue.

1. The authors should describe parameterizations and modification of model more concretely. Especially, following the parts:

"we have calibrated the parameters of fire spread, fuel combustibility, 180 and carbon combustion to reproduce the observed magnitude and temporal variations of burned area and carbon emission in satellite data."

How did the authors calibrate?

R: In the latest version, we used the crop fraction from LUH2, which is updated annually from 1948-2014, to exclude fire in the croplands. The crop fraction in LUH2 is much smaller than the crop data GLC2000 (a constant crop fraction at the year 2000) used in Huang et al. (2020).

With a smaller crop fraction data, we have reduced the fire spread rate and fuel combustibility to produce a similar burned area as before. Besides, the carbon emission per burned area fraction ($\frac{carbon\ emission}{Burned\ area\ fraction}$) in Southern Africa (Huang et al. 2020) is larger than the observed value from GFED4s. We have decreased the leaf combustion completeness accordingly. The above description is added in Lines 185-188.

"we optimize photosynthesis-related parameters according to the observed GPP magnitude in both wet seasons and dry seasons as follows."

How did the authors optimize?

R: The previous version SSiB4/TRIFFID-Fire overestimates vegetation productivity (GPP) in the dry season and fails to capture its seasonal variations. In the updated version, we have decreased the root-zone soil moisture potential factor ($f(\theta)$) in dry months which reflects the impacts of soil water deficit on transpiration. The whole process of calibration is discussed in the following paragraph. We have revised the last sentence to avoid confusion: "To reduce GPP magnitude in dry seasons, we optimized root-zone soil moisture potential factor f(θ), a parameter that determines transpiration in SSiB4/TRIFFID-Fire, to constrain photosynthesis activities. The procedure is introduced as follows."

"Therefore, we adjusted the 205 coefficients c1 and c2 for C4 grasses to reflect the effects of soil water deficit on transpiration in a wider range of soil moisture between 0.3 – 0.6 (Fig. 2a). ð   (ð    ) for C3 grasses is also adjusted but is designed to be less sensitive to low moisture conditions (compared to C4 grasses) to make it more adaptive in the dry area (Fig. 2b)."

How did the authors adjust?

R: In the equation, $c_1$ represents the wilting point at which stomates close completely, and $c_2$ is a slope factor controlling how sensitive the vegetation responds to soil water deficit. For both C3 and C4 grasses, we have decreased $c_2$ to reduce the slope so that the vegetation responds to water deficit at relatively wetter conditions (Fig. 2 original and calibrated lines). The $c_1$ for C3 grasses is set to be smaller than that for C4 grasses so that its stomates close at a drier condition and it is more sustainable to drought than C4 grasses. The above description is added in Lines 213-218

2. I think it is better to compare the results of improved model and those of previous model and show how the new model became better.

R: We agree that it is necessary to compare the updated model with the previous one to show the improvements. As shown in Fig. R1 (also added to supplementary material Fig. S1), the previous SSiB4/TRIFFID-Fire overestimates GPP in the dry season compared to observations and does not capture its seasonal cycle well. The improved version has largely decreased GPP in dry months with the temporal correlation coefficient increased to 0.91. The reduction in GPP also decelerates vegetation recovery in the dry season, making it more consistent with field observations. Therefore, the improvement is vital to study fire effects on vegetation and surface energy.

[Figure]

Fig. R1 Monthly GPP (g C m-2 mon-1) in model and observation (a) before calibration and (b) after calibration

The point-by-point monthly correlations are improved in both GPP and LAI (Fig. R2). In previous model results, most C3 and C4 dominant areas (5-20 °S) have a temporal correlation of 0.4-0.6 between modeled GPP and observed GPP from FLUXNET-MTE

(Figs. R2a). With model improvement, the seasonal variations in these regions have been largely improved with a correlation coefficient between 0.5-0.7 (Figs. R2b). A similar conclusion can be drawn from LAI, which has even larger improvements compared to GPP. We add Figs. R1 and R2 to supplementary material and the above discussion to Lines 323 to 329.

[Figure]

Figure R2 (a-b) Point-by-point climatology monthly correlation between FLUXNET-MTE and SSiB4/TRIFFID-Fire GPP (a) before calibration and (b) after calibration. (c-d) Point-by-point monthly correlation between GLASS and SSiB4/TRIFFID-Fire LAI (c) before calibration and (d) after calibration.